# Tailoring Confining Jacket for Concrete Column Using Ultra High Performance-Fiber Reinforced Cementitious Composites (UHP-FRCC) with High Volume Fly Ash (HVFA)

**DOI:** 10.3390/ma12234010

**Published:** 2019-12-03

**Authors:** Alessandro P. Fantilli, Lucia Paternesi Meloni, Tomoya Nishiwaki, Go Igarashi

**Affiliations:** 1Department of Structural, Geotechnical and Building Engineering, Politecnico di Torino, 10129 Torino, Italy; s238354@studenti.polito.it; 2Department of Architecture and Building Science, Tohoku University, Sendai 980-8579, Japan; tomoya.nishiwaki.e8@tohoku.ac.jp (T.N.); go.igarashi@concrete.t.u-tokyo.ac.jp (G.I.); 3Department of Civil Engineering, The University of Tokyo, Tokyo 113-8656, Japan

**Keywords:** high volume fly ash (HVFA), steel reinforcing fiber, jacketing, carbon footprint, substitution strategy, environmental impact

## Abstract

Ultra-High Performance Fibre-Reinforced Cementitious Composites (UHP-FRCC) show excellent mechanical performances in terms of strength, ductility, and durability. Therefore, these cementitious materials have been successfully used for repairing, strengthening, and seismic retrofitting of old structures. However, UHP-FRCCs are not always environmental friendly products, especially in terms of the initial cost, due to the large quantity of cement that is contained in the mixture. Different rates of fly ash substitute herein part of the cement, and the new UHP-FRCCs are used to retrofit concrete columns to overcome this problem. To simulate the mechanical response of these columns, cylindrical specimens, which are made of normal concrete and reinforced with different UHP-FRCC jackets, are tested in uniaxial compression. Relationships between the size of the jacket, the percentage of cement replaced by fly ash, and the strength of the columns are measured and analyzed by means of the eco-mechanical approach. As a result, a replacement of approximately 50% of cement with fly ash, and a suitable thickness of the UHP-FRCC jacket, might ensure the lowest environmental impact without compromising the mechanical performances.

## 1. Introduction

In the last decades, Ultra-High Performance Fiber-Reinforced Cementitious Composites (UHP-FRCC) have been developed to meet the requests of the construction industry [1,2,3]. UHP-FRCCs can enhance the resistance of buildings and infrastructures due to the ultra-high strength, high ductility, durability, and energy absorption capacity, when compared with normal strength concrete or traditional FRCC. In fact, UHP-FRCCs show a compressive strength that was larger than 150 MPa, combined with high tensile and flexural strengths. Such performances are achieved with a low water/binder ratio, high content of cementitious materials, and by incorporating a copious amount of fibers (steel, polymeric, glass, etc.) [4,5,6,7].

One of the most well-known and relevant applications of UHP-FRCC is the retrofitting of existing structures, especially the jacketing of concrete columns and beams, due to these mechanical properties. The aim is to harden those parts of the existing structures that are exposed to high environmental and mechanical actions, especially in the most highly stressed cross-sections and in the structural joints [8,9,10].

However, this high-performance material has, in parallel, high environmental impact, because of the high content of cement in the mixture [3,5,6,11,12]. Indeed, it is unanimously accepted that the compressive strength of concrete is in direct proportion with the power 2 of the cement content and, consequently, this strength is proportional to the power 2 of the CO_2_ emission per cubic meter of concrete. Therefore, the “material substitution strategy” that Habert and Roussel introduced [11] is generally adopted. As the name suggests, it consists of a partial replacement of cement with supplementary cementitious materials, in order to reduce the environmental impact. Specifically, different amounts of cement are substituted by fly ash, a waste by-product that is derived from coal burning. When the mass of fly ash is higher than 50% of the total cementitious materials, the concrete system takes the name of High Volume Fly Ash (HVFA) [13]. It must be remarked that the substitution of cement with fly ash might not always be beneficial. HVFA significantly reduces the environmental impact, but it also leads to a decrease of concrete strength, especially in the UHP-FRCC [14,15].

Some researches, regarding the high strength concrete and UHP-FRC using high quantity of admixtures from by-products, were performed in the last years [16,17,18]. On the other hand, most of these studies only focused on the development of the cement-based material, without focusing on the structural applications. Accordingly, a new experimental campaign has been carried out on normal-strength concrete cylinders that were reinforced with UHP-FRCC jackets, with the aim of simulating the confinement effect in columns. Different jackets are tailored to investigate the relationship between the thickness of the reinforcing layer and the mechanical performances of the reinforced column. In some of them, different percentages of fly ash substitute the cement for achieving the best mechanical and ecological performances [19]. A design procedure is also proposed to select the best solution by defining the optimal replacement rate of cement with fly ash.

## 2. Experimental Investigation: Materials and Methods

Figure 1 illustrates the cylindrical sample subjected to uniaxial compression. It consists of a normal-strength concrete core, with a radius *r*_0_ = 50 mm and a height *H* = 200 mm, confined by a UHP-FRCC jacket of *H*_1_ = 178 mm and different thickness (*t*_i_ = 25, 37.5, 50, and 75 mm). As no standard exists to test the confinement effects, the minimum thickness of 25 mm was determined due to the limitation of the casting procedures of the UHP-FRCC layer into a narrow gap.

According to the mix proportion that is shown in Table 1, the concrete cores have been made with High Early Strength Portland Cement (HESP; Density: 3.14 g/cm^3^, Specific surface area: 4490 cm^2^/g Ignition loss: 1.08%), the combination of land sand and crushed sand as fine aggregates (S), crushed stone as coarse aggregates (G), tap water (W), and superplasticizer (SP_1_; Polycarboxylate-based, Density: 1.03 g/cm^3^).

The UHP-FRCCs used herein are made by the following materials [3,13,14]:
Low Heat Portland Cement (LHC; Density: 3.24 g/cm^3^, Specific surface area: 3640 cm^2^/g, C_2_S: 57%, C_3_A: 3%, MgO: 0.6%, SO_3_: 2.78%, Ignition loss: 0.72%)Undensified Silica Fume (SF; Density: 2.20 g/cm^3^, Bulk density: 0.20–0.35 g/cm^3^, Coarse particles >45 µm: less than 1.5, SiO_2_: more than 90%, Ignition loss: less than 3.0%)Fly Ash (FA; Density: 2.31 g/cm^3^, Specific surface area: 4050 cm^2^/g, Coarse particles >45 µm: 5%, SiO_2_: 54.8%, Ignition loss: 1.2%)Silica sand (Ss; Density: 2.60 g/cm^3^, Average particle size: 0.212 mm, SiO_2_: 98.49%, Al_2_O_3_: 0.49%)Wollastonite mineral fibers (Wo; CaSiO_3_, Density: 2.60 g/cm^3^, Length: 50–2000 μm, Aspect ratio: 3–20, SiO_2_: 49.71%, CaO: 45.87%, Ignition loss: 1.94%)Specific superplasticizer (SP_2_; Polycarboxylate-based, Density: 1.05 g/cm^3^, Solid part: 30%)De-foaming Agent (DA; Density: 1.05 g/cm^3^)Steel micro-fibers (OL—1% in volume), with a length of 6 mm (see Figure 2a)Steel macro-fibers (HDR—1.5% in volume), with a length of 30 mm (see Figure 2b)


Following the material substitution strategy, four mixtures have been tailored for the UHP-FRCC jackets, modifying only the percentage of cement and fly ash. Table 2 shows the mix proportion of the four series of UHP-FRCC (FA0, FA20, FA50, and FA70), with, respectively, 0%, 20%, 50%, and 70% of cement replaced by fly ash. In this Table, the components of each series are reported as a percentage with respect to the weight of the binder, whereas Table 3 shows the mix proportion of UHP-FRCC jackets in kg/m^3^. In each series, the water-binder ratio is constant and equal to 0.16, because it guarantees a good balance between the flowability properties and the strength of the hardened concrete, according to the authors’ previous study [3]. Note that SP_2_ contains 30% of the solid part, which is taken into account to calculate the water-binder ratio. The flow table tests were performed in accordance with the Japanese Standard JIS R 5201 [20], which complies with ASTM C 1437 [21]. In particular, for FA0, FA20, FA50, and FA70, a diameter of 190 mm, 260 mm, 290 mm, and 250 mm, was, respectively, measured in the tests. Having these consistencies, all of the UHP-FRCs were cast and compacted in the jacket mould without any segregation.

Ten days after casting the concrete cores (which simulate herein an existing column), the UHP-FRCC jackets are cast around using paper cylinders with a height of 178 mm as a disposable formwork, as the proposed jacket has to be applied to existing structures.

Subsequently, the paper formworks were removed two days later and the specimens were subjected to steam curing for the following 48 h. In this way, the hydration process of the binder was accelerated, especially in the UHP-FRCC jackets containing large amounts of fly ash. As depicted in Figure 3, the temperature was slowly increased at a rate of 15 °C per hour up to 90 °C to avoid any crack generation produced by the thermal gradient. Uniaxial compression tests are performed after 28 days from the concrete core casting. A universal testing machine (UTM) with a maximum capacity of 1000 kN is used to apply the compressive load by moving the displacement of the stroke (of the loading cell) at a constant velocity of less than 0.3 mm/min. external strain gauges and embedded strain gauges were used to measure the strain during the tests.

Four strain gauges are applied on the jacket’s surface: two in the vertical direction and two in the horizontal direction. The vertical gauges measure the strain along the longitudinal axis, while those horizontally oriented gauges measure the swelling of the specimen. Embedded strain gauges were installed inside in the middle of the moulds before casting the concrete cores to record the vertical strain of the concrete core during the compression test.

As shown in Table 4, for some of the thickness of the jacket, three samples are tested for each mixture. In addition, for each series, one concrete core with no reinforcement is tested to estimate the mechanical properties without UHP-FRCC jackets.

## 3. Results

### 3.1. Mechanical Performances

Table 5 shows the mechanical properties of the UHP-FRCC used in the jackets. The compressive strength and the elastic modulus both do not change with age because of the acceleration of hydration due to the steam curing. Table 5 summarizes the results of the uniaxial compression tests on the jacketed specimens. In this table, the average values of the maximum load (P_max_)—and of the compressive strength (σ_max_) as well, the Young Modulus (*E*_cm_) and the Poisson’s Ratio (ν) of the composite specimens are reported. *E*_cm_ and ν were both calculated at one-third of the maximum load, according to the Japanese Standard JIS A 1149 [22], which complies with ISO 6784 [23]. In Table 6, the strength is also normalized with respect to the strength of the unconfined concrete cylinders (i.e., f_c_CORE_). *E*_cm_ increased with the thickness of the jacket due to the confined effect. In the same manner, ν decreased as the jacket thickness increased. The tests on the specimen that is shown in Figure 1 were performed until the failure, which is generally produced by the formation of a large tensile crack in the UHP-FRCC jacket (see the right edge photo of Figure 3). During the first stage of loading, the jacked was uncracked and a linear relationship between the stress (calculated by dividing the load by the cross-sectional area of the core concrete) and the strain can be observed (see Figure 4). When the first crack appeared in the jacket, the slope of this relationship drastically reduces in all of the specimens. This is due to the fact that multiple fine cracks, having a width lower than 0.1 mm, formed in the jacket. Despite the growing number of cracks, the stress continuously increased up to peak, where the tensile strains localised in a single crack of the jacket and the failure occurred.

The composition of the UHP-FRCC binder considerably influences the mechanical performance of the column. In particular, the effect of the substitution strategy on the mechanical properties is evident in Figure 4a, which reports the stress-strain curves of the concrete cylinders that were confined with a jacket of 25 mm, but with 0%, 20%, 50%, and 70% of cement being replaced by fly ash. The results that were obtained in the case of the FA20 mixture (σ_max_ = 59.06 MPa) are close to those that were achieved from the FA0 mixture (σ_max_ = 61.30 MPa). This is the same tendency of the results of UHP-FRCC material itself, as shown in Table 5. In other words, the replacement of 20% in weight of cement content is paid with a loss of 4% of the maximum compressive strength. When considering the cylinder confined with a jacked of FA0 and *t*_i_ = 25 mm as a reference, the decrement of the compressive strength is about 14% and 18% when 50% and 70% of cement is replaced by fly ash, respectively.

On the other hand, Figure 4b reports the average stress-strain curves of the cylinders that were reinforced with the same type of jacket (FA20), but of different thickness (i.e., 25, 50, 37.5, and 70 mm). In this case, the strength increases with the thickness of the reinforcing UHP-FRCC layer.

For each UHP-FRCC series, Figure 5a shows the relationship between the thickness of the jacket and the strength of the reinforced cylinders. In all of the cases, the following linear relationship can be used to predict the compressive strength:
(1)$$\sigma_{\max}=s\cdot t_{i}+f_{c\_\mathrm{CORE}}$$
where *s* = slope of the linear relationship. The coefficient *s* can be separately computed for each series, as shown in Figure 5b, and the values can be plotted as a function of the replacement rate of cement with fly ash (see Figure 6). The slope gradually reduces as the percentage of substitution of cement increases (*C_sub_*). Thus, the following linear correlation can be introduced:
(2)$$s=-0.004\cdot C_{sub}+0.584$$
where *s* is measured in MPa/mm.

### 3.2. Ecological Performances

The parameter considered evaluating the ecological performances is the amount of CO_2_ related to the production of 1 m^3^ of UHP-FRCC. The amount of CO_2_ emitted per unit volume of each mixture is calculated in accordance with the inventory analysis [24] while using the values that were provided by the Japan Concrete Institute (JCI) [25], which listed the main materials used in cementitious composites and the relative carbon footprint. Such values are reported in Table 7 in terms of kg of CO_2_ released in the atmosphere for the production of one-ton of material.

Figure 7 compares the environmental impacts of the various jackets, which were obtained by multiplying the values reported in Table 7 and the mass of materials used to cast the four series of UHP-FRCC. For the sake of completeness, Figure 7 also shows the values of some specimens that has not been tested. In this Figure, when the amount of cement replaced with fly ash is quite high (>50%), the environmental impact is considerably reduced with respect to FA0 (it is halved for FA70). Moreover, it can be noticed that a reduced environmental impact is attained by decreasing the thickness of the jacket and, in parallel, by increasing the percentage of fly ash in the mixture. Here, the CO_2_ emission due to the steam curing is not taken into account, because all of the series were subjected to the same procedures and, subsequently, a comparative analysis among all of the specimens is performed.

### 3.3. Eco-Mechanical Analysis and Design Procedure

By means of the approach that was proposed by Fantilli and Chiaia [19], ecological and mechanical analyses can be combined to define the best material. Compressive strength (σ_max_) is considered as the functional unit, herein called the mechanical index (MI), whereas the ecological impact is evaluated through the carbon footprint (ecological index—EI). The reference values MI_inf_ (i.e., the minimum mechanical performance) and EI_sup_ (i.e., the maximum impact) are those of the concrete cylinder reinforced with the UHP-FRCC jacket without the substitution of cement with fly ash (i.e., FA0) and with a thickness *t*_i_ = 25 mm.

Figure 8 shows the non-dimensional chart that was used to perform the comparative analysis among the concrete specimens reinforced with the UHP-FRCC layer. In this diagram, the ratios MI/MI_inf_ and EI_sup_/EI are the abscissa and the ordinate, respectively. In other words, the following formula are used to define the non-dimensional axes in Figure 8:
(3)$$\frac{\mathrm{MI}}{{\mathrm{MI}}_{\inf}}=\frac{\sigma_{\max}}{\sigma_{\max}\;of\;FA0\;with\;t_{i}=25\;\mathrm{mm}}$$
(4)$$\frac{{\mathrm{EI}}_{\sup}}{\mathrm{EI}}=\frac{\mathrm{kg}\;{\mathrm{CO}}_{2}\;of\;FA0\;with\;t_{i}=25\;\mathrm{mm}}{\mathrm{kg}\;{\mathrm{CO}}_{2}}$$


Most of the experimental results fall in Zone 2, where the mechanical performances are increased at the expense of an environmental impact higher than that of FA0 with *t*_i_ = 25 mm (for which MI/MI_inf_ = EI_sup_/EI = 1).

However, a group of experimental values falls within Zone 4, which shows ecological performances greater than those shown by the reference specimen, but lower mechanical performances. Although none of the test results fall within Zone 3, where the ecological and mechanical performances are both improved, an area of the possible best solutions can be defined in Figure 8. More precisely, the UHP-FRCC jackets with a thickness between 37.5–50 mm and made with FA70 series, and those of the FA50 series with thickness being included in the range 25–50 mm (see the dashed lines), might perform better than the reference cylinders (i.e., part of the dashed lines falls within Zone 3). The same is also valid for some thickness (within the range 25~37.5 mm) of the series FA20, as evidenced by the dashed line that is reported in Figure 8.

Nevertheless, Equations (1) and (2) can be used to relate the compressive stress, the thickness of the jacket, and the type of fiber-reinforced concrete to design the exact UHP-FRCC layer. In particular, the design procedure that is shown in Figure 9 can be introduced with the aim of increasing the strength of the UHP-FRCC jacketing system, and reducing the CO_2_ emissions as well.

As a result, Figure 10 shows the curves that relate the CO_2_ emissions with the percentage of cement replaced by fly ash for three values of the compressive strength σ_max_ (i.e., 55, 65, and 75 MPa). For each load carrying capacity, four thicknesses of the jackets have been obtained referring to the mixtures FA0, FA20, FA50, and FA70.

If σ_max_ ≤ 55 MPa is required, the CO_2_ emission does not change with the thickness or the percentage of cement replacement with fly ash. Conversely, as the required strength increases, the corresponding CO_2_ emission has a minimum in correspondence of a specific thickness. For instance, the optimal replacement of cement with fly ash to achieve a compressive stress of 75 MPa is 50% and the corresponding thickness is 67 mm. In the three curves that are shown in Figure 10, the best substation rate is at 50%, as obtained by Fantilli et al. [26] in the reinforced concrete beam. Indeed, for lower substitution rates, thickness reduces, but the CO_2_ emissions are higher.

When the substitution strategy is forced to higher percentages, the same mechanical performances can only be obtained through solutions with a high environmental impact, as the thickness of the jacket increases. In other words, the UHP-FRCC jacket with a low fly ash content shows mechanical performances that are not compensated by the environmental impact. In the same way, the jacketing system is so thick to produce a large environmental impact for large substitutions of cement.

## 4. Conclusions

According to the results of the tests previously described, the following conclusions are drawn:
A linear relationship between the thickness of the jacket and the compressive strength of concrete columns has been found. By means of this relationship, the thickness of the UHP-FRCC jacket might be adjusted to achieve the desired mechanical performances.The strength of the confined column linearly decreases as the substitution of cement with fly ash increased in the UHP-FRCC jacket.Through the eco-mechanical analysis, it is possible to demonstrate that the partial replacement of cement with fly ash, combined with a suitable thickness of the jacket, might simultaneously guarantee the best mechanical and ecological performances.A design procedure has been introduced to optimise the UHP-FRCC jacketing system and reduce the environmental impact as well. Particularly, for a given strength of the column, the impact of the UHP-FRCC jacket has a minimum in correspondence of a substitution rate of cement with fly ash that is close to 50%.


## Figures and Tables

**Figure 1 materials-12-04010-f001:**
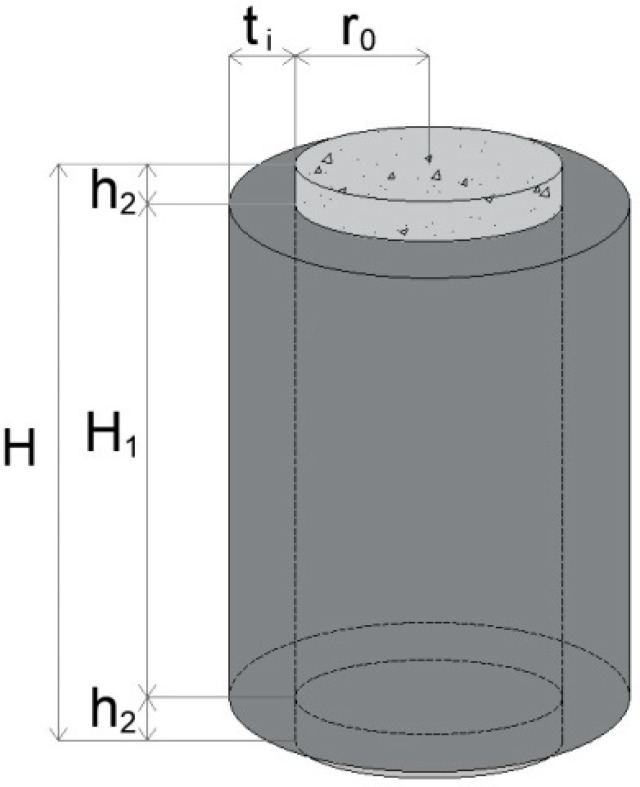
Geometrical properties of the specimens.

**Figure 2 materials-12-04010-f002:**
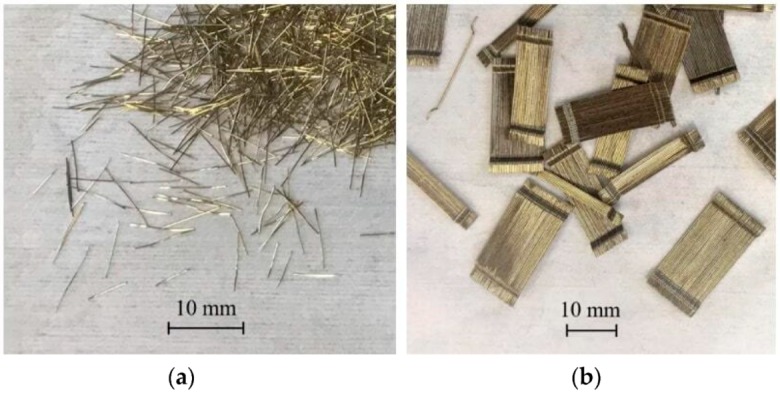
Steel Micro-fibers OL (**a**) and steel Macro-Fibers HDR (**b**) used to reinforce the Ultra-High Performance Fibre-Reinforced Cementitious Composites (UHP-FRCC) layers.

**Figure 3 materials-12-04010-f003:**
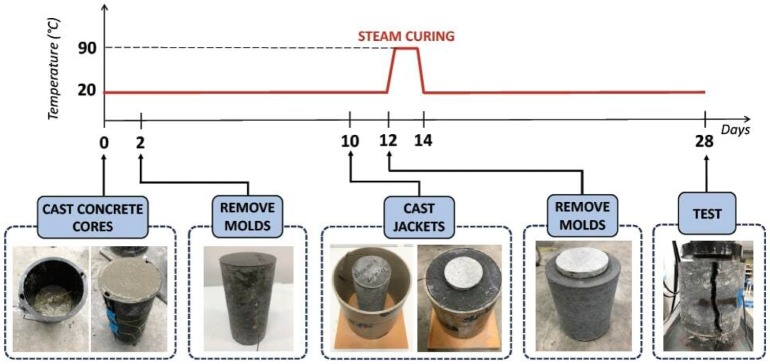
The preparation of the specimens.

**Figure 4 materials-12-04010-f004:**
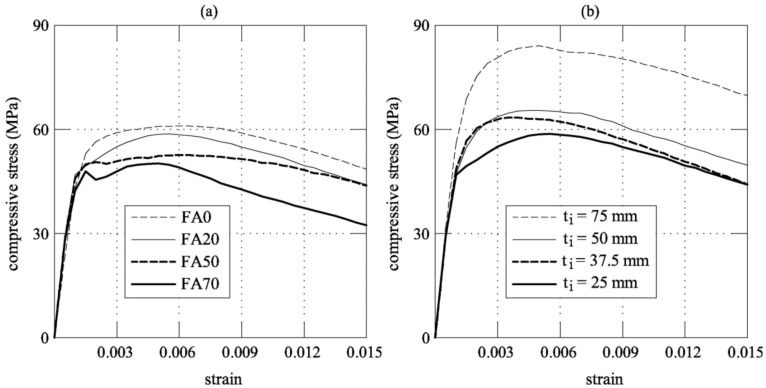
Stress-strain curves of the jacketed cylinders: (**a**) behaviour of different UHP-FRCC jackets having a constant of thickness *t*_i_ = 25 mm; (**b**) behaviour of the same UHP-FRCC jacket (FA20) having different thickness.

**Figure 5 materials-12-04010-f005:**
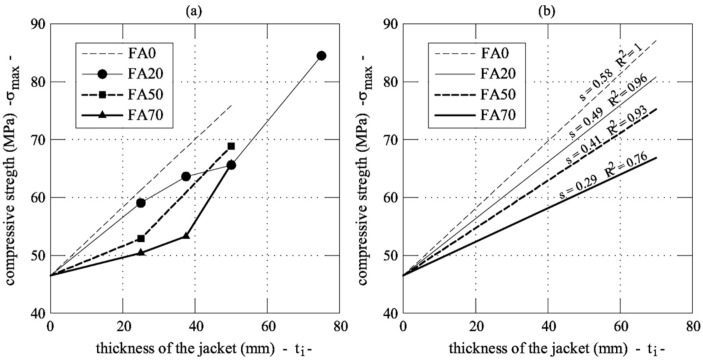
Compressive strength vs. thickness of jacket in the four ultra high performance fiber reinforced concrete (UHP-FRC) series investigated herein: (**a**) results from the tests; and, (**b**) the trend lines of the experimental data.

**Figure 6 materials-12-04010-f006:**
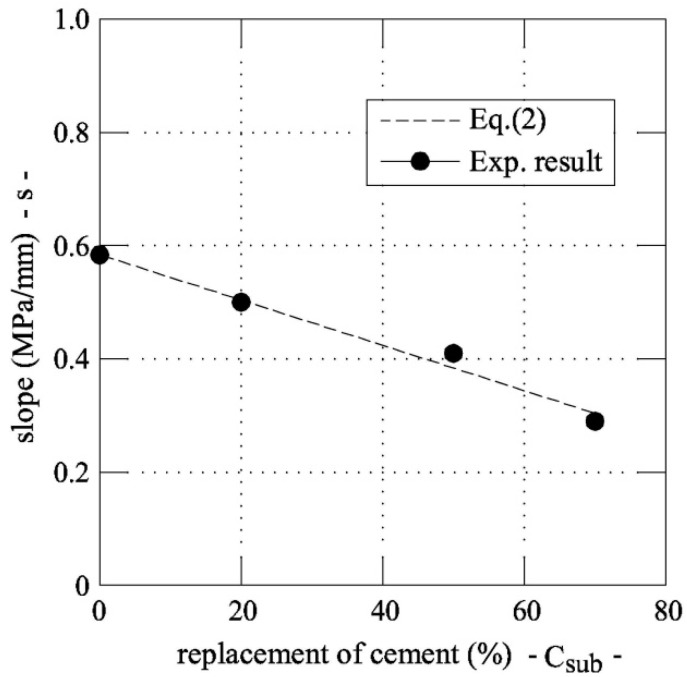
Formula for predicting the slope of the linear approximation of Equation (1).

**Figure 7 materials-12-04010-f007:**
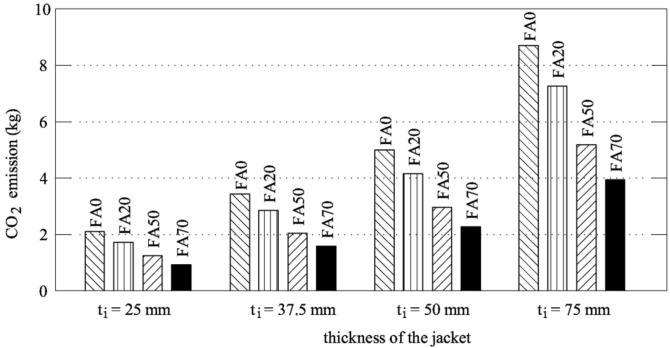
The ecological performances of the UHP-FRCC jackets used to reinforce concrete cores.

**Figure 8 materials-12-04010-f008:**
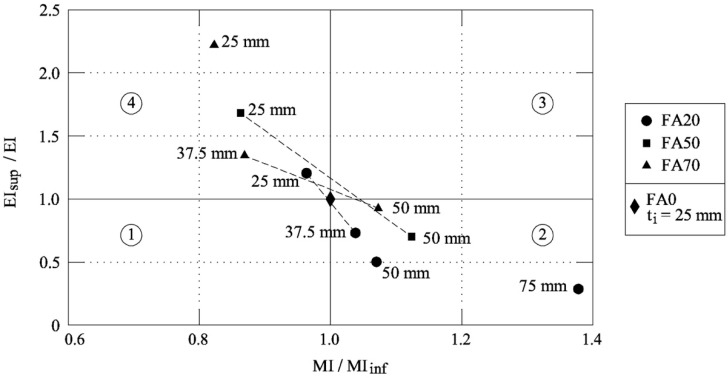
Eco-Mechanical analysis of the UHP-FRCC jackets [19].

**Figure 9 materials-12-04010-f009:**
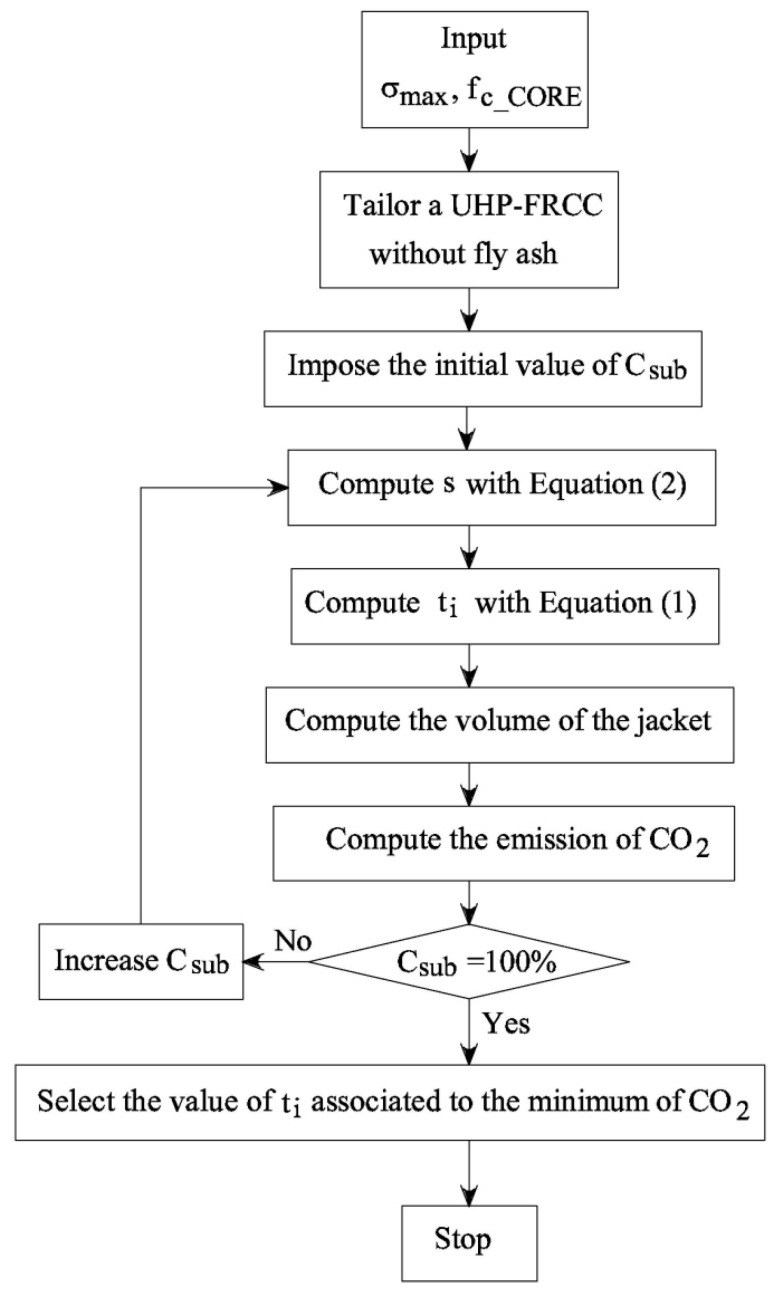
The design procedure used to optimize the UHP-FRCC jacket of the concrete columns.

**Figure 10 materials-12-04010-f010:**
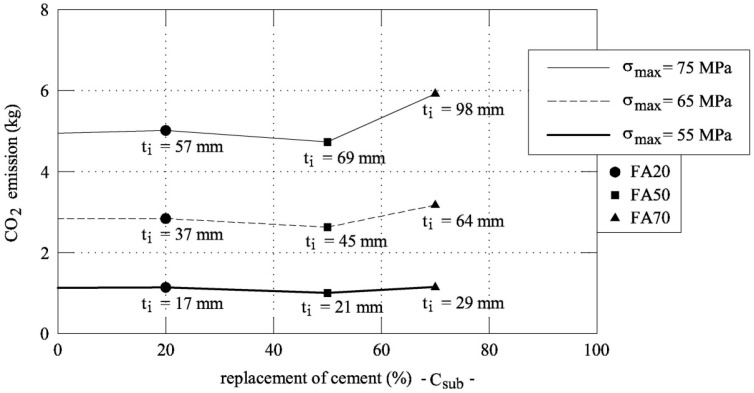
The ecological impact of UHP-FRCC jackets made with different mixtures (having different σ_max_).

**Table 1 materials-12-04010-t001:** Concrete mixture used to cast the cores.

HESP (kg/m^3^)	S (kg/m^3^)	G (kg/m^3^)	W (kg/m^3^)	SP_1_ (kg/m^3^)
300.3	836	900.3	171.7	1.7

**Table 2 materials-12-04010-t002:** Mix proportion of the UHP-FRCC (in weight %) referred to the binder (B).

Series	Binder (B)			Ss/B	Wo/B	W/B	SP_2_/B	DA/B
	LH C/B	FA/B	SF/B					
FA0	82		18	35	13	14.5	2.2	0.02
FA20	65.6	16.4	18	35	13	14.3	2.6	0.02
FA50	41	41	18	35	13	14.3	2.6	0.02
FA70	24.6	57.4	18	35	13	14.3	2.6	0.02

**Table 3 materials-12-04010-t003:** Mix proportion of UHP-FRCC jackets in kg/m^3.^

Series	LHC (kg/m^3^)	FA (kg/m^3^)	SF (kg/m^3^)	Ss (kg/m^3^)	Wo (kg/m^3^)	W (kg/m^3^)	SP_2_ (kg/m^3^)	DA (kg/m^3^)
FA0	1217.52		267.26	519.67	193.02	214.70	32.67	0.30
FA20	939.24	234.81	257.72	501.12	186.13	204.74	37.23	0.29
FA50	558.33	558.33	245.12	476.63	177.03	194.74	35.41	0.27
FA70	324.43	757.00	237.39	461.59	171.45	188.59	34.29	0.26

**Table 4 materials-12-04010-t004:** List of specimens tested in uniaxial compression.

Series	Thickness of the UHP-FRCC Jacket			
	25 (mm)	37.5 (mm)	50 (mm)	75 (mm)
FA0	3 specimens		3 specimens	
FA20	3 specimens	3 specimens	3 specimens	3 specimens
FA50	3 specimens		3 specimens	
FA70	3 specimens	3 specimens	3 specimens	

**Table 5 materials-12-04010-t005:** The compressive strength and Young’s modulus of UHP-FRCC used in the jackets.

Series	Age	Compressive Strength (MPa)	Young’ Modulus (GPa)
FA0	1 week	193.8	46.34
	4 weeks	197.93	45.90
FA20	1 week	193.31	43.39
	4 weeks	179.65	43.38
FA50	1 week	146.97	36.20
	4 weeks	154.55	37.66
FA70	1 week	121.50	33.40
	4 weeks	121.98	34.12

**Table 6 materials-12-04010-t006:** The average values of the parameters measured in the uniaxial compression tests.

Series	Jacket (mm)	P_max_ (kN)	σ_max_ (MPa)	σ/f_c_CORE_	*E*_cm_ (MPa)	ν
FA0	25	481.47	61.30	1.32	37.90	0.120
	50	595.93	75.88	1.63	40.40	0.104
FA20	25	463.86	59.06	1.27	37.48	0.179
	37.5	499.80	63.64	1.37	38.30	0.157
	50	515.27	65.61	1.41	38.67	0.125
	75	663.53	84.48	1.82	41.70	0.121
FA50	25	415.40	52.89	1.14	36.25	0.195
	50	541.00	68.88	1.48	39.23	0.162
FA70	25	396.07	50.43	1.08	35.74	0.216
	37.5	418.40	53.27	1.15	36.66	0.168
	50	516.60	65.78	1.41	38.70	0.123
Unconfined cylinders		356.37	46.52	1		

**Table 7 materials-12-04010-t007:** CO_2_ emissions of UHP-FRCC components [25].

Components	kg of CO_2_/t
LHC	769
FA	29
Sand	4.9
Water	34.8
SP	150
Fibers	1320

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
