# Peer review of "Tailoring Confining Jacket for Concrete Column Using Ultra High Performance-Fiber Reinforced Cementitious Composites (UHP-FRCC) with High Volume Fly Ash (HVFA)"

_materials, 2019, doi:10.3390/ma12234010_

Round 1

Reviewer 1 Report

Title:  Tailoring confining jacket for concrete column using ultra high performance-fiber reinforced cementitious composites (UHP-FRCC) with high volume fly ash (HVFA)

This paper investigates the feasibility of using UHP-FRCC with high volume fly ash for retrofitting of normal concrete column. The ecological performances of UHP-FRCC are evaluated and the   design procedure to optimize the UHP-FRCC jacketing system is introduced.  The topic fit with the scope of the journal.  The paper can be accepted for its publication in the journal after major revisions. 

Please consider the following comments and suggestions.

1 There are several published papers about the utilizing of fly ash to replace cement in UHPC, the related papers for the authors are as follows. Please clarify the new UHP-FRCCs and the main contrition of this study in Abstract.  

[1] R. Yu, P. Spiesz, H.J.H. Brouwers. Development of an eco-friendly Ultra-High Performance Concrete (UHPC) with efficient cement and mineral admixtures uses, Cement and Concrete Composites, Volume 55, 2015, Pages 383-394.

[2] Parham Aghdasi, Claudia P. Ostertag. Green ultra-high performance fiber-reinforced concrete (G-UHP-FRC), Construction and Building Materials, Volume 190, 2018, Pages 246-254.

[3] ÇaÄŸlar Yalçınkaya, Halit Yazıcı. Effects of ambient temperature and relative humidity on early-age shrinkage of UHPC with high-volume mineral admixtures, Construction and Building Materials, Volume 144, 2017, Pages 252-259.

2 In the section 2 Experimental investigation: materials and methods, what is the basis of the design of specimens in Fig. 1?  Does the test method for confinement effect from the standard?

3 In the section 2 Experimental investigation: materials and methods, the selected thickness (ti) of UHP-FRCC seems inappropriate. When the UHP-FRCC is employed to retrofit the existing structure, the thickness of it is thin because of the ultra-high strength, high ductility and durability of UHP-FRCC. In Fig.1, the radius of normal-strength concrete core is 50mm. However, the maximum thickness of UHP-FRCC is 75mm which may not be reasonable. Please explain the selected thickness of UHP-FRCC.

4 In Table 2, why the W/B of FA0 is different from other series? In Table 3, the content of raw materials of UHP-FRCC, except the LHC and FA, are also different in all series, please explain.

5 In Figure 3, the UHP-FRCC jackets are cast around the cores at ten days, is there any reason for it?

6 In Table 4, the four types of thickness of the UHP-FRCC jacket should be tested for all series, not just in FA20. Because, the equations (1) and (2) are obtained from these test data, the precision of the relationship for equations (1) and (2) can be ensured in the presence of enough data.

7 The flowability of all series should be provided. In general, the particle size of fly ash is not same with that of the cement, which can be influenced the flowability of UHP-FRCC.

8 In section 3.1, the brief analysis about the Young modulus and the Poisson’s ratio should be added.

9 In Figure 5, the goodness of fit between test results and trend lines should be provided.

10 In Figure 8, please provide more evidences and discussions about the defined possible best solutions and the dashed lines. The dashed line between 37.5 mm-75 mm in FA20 seemed also falls within Zone 3.

Author Response

1

Reviewer’s comments

1 There are several published papers about the utilizing of fly ash to replace cement in UHPC, the related papers for the authors are as follows. Please clarify the new UHP-FRCCs and the main contrition of this study in Abstract.

[1] R. Yu, P. Spiesz, H.J.H. Brouwers. Development of an eco-friendly Ultra-High Performance Concrete (UHPC) with efficient cement and mineral admixtures uses, Cement and Concrete Composites, Volume 55, 2015, Pages 383-394.

[2] Parham Aghdasi, Claudia P. Ostertag. Green ultra-high performance fiber-reinforced concrete (G-UHP-FRC), Construction and Building Materials, Volume 190, 2018, Pages 246-254.

[3] ÇaÄŸlar Yalçınkaya, Halit Yazıcı. Effects of ambient temperature and relative humidity on early-age shrinkage of UHPC with high-volume mineral admixtures, Construction and Building Materials, Volume 144, 2017, Pages 252-259.

Authors’ answer

According to the reviewer’s suggestion, other papers were added in the reference list and some comments were also reported in the introduction. In particular, the proposed approach to tailoring new UHP-FRC UHP-FRC with HVFA is strictly connected to the practical application (i.e., the confined columns). Thus, the following sentences were added at:

Page 2, lines 11-14

Some researches, regarding high strength concrete and UHP-FRC using high quantity of admixtures from by-products, were performed in the last years [16-18]. On the other hand, most of these studies only focused on the development of the cement-based material, without focusing on the structural applications.

2

Reviewer’s comments

2 In the section 2 Experimental investigation: materials and methods, what is the basis of the design of specimens in Fig. 1? Does the test method for confinement effect from the standard?

3 In the section 2 Experimental investigation: materials and methods, the selected thickness (ti) of UHP-FRCC seems inappropriate. When the UHP-FRCC is employed to retrofit the existing structure, the thickness of it is thin because of the ultra-high strength, high ductility and durability of UHP-FRCC. In Fig.1, the radius of normal-strength concrete core is 50mm. However, the maximum thickness of UHP-FRCC is 75mm which may not be reasonable. Please explain the selected thickness of UHP-FRCC.

Authors’ answer

As an novel approach is described in this manuscript, there is no standard for measuring the effect of confinement by using UHP-FRCC layers. We just reproduced in scale the confinement of an existing concrete column. Hence, the minimum thickness of 25 mm was determined by the casting the UHP-FRCC layer into the narrow gap. The following lines have been added to explain the choice of the thickness:

Page 2, lines 24-26

As no standard exists to test the confinement effects, the minimum thickness of 25 mm was determined due to the limitation of casting procedures of the UHP-FRCC layer into a narrow gap.

3

Reviewer’s comments

4 In Table 2, why the W/B of FA0 is different from other series? In Table 3, the content of raw materials of UHP-FRCC, except the LHC and FA, are also different in all series, please explain.

Authors’ answer

The water-binder ratio is almost constant of 0.16. However, the slight variation reported in table 3 is due to the solid parts contained in SP2. This solid part was eliminated from the calculation of water-binder ratio. This is remarked at:

Page 3, lines 26-29

In each series, the water-binder ratio is constant and equal to 0.16, because it guarantees a good balance between the flowability properties and the strength of the hardened concrete, according to the previous study of the authors [3]. Note that SP2 contains 30% of the solid part, which is taken into account to calculate the water-binder ratio.

4

Reviewer’s comments

5 In Figure 3, the UHP-FRCC jackets are cast around the cores at ten days, is there any reason for it?

Authors’ answer

There is not a particular reason for this way of casting. We just want to simulate the jacketing of an existing column (i.e. the concrete core), as remarked as:

Page 3., lines 34-36

As the proposed jacket has to be applied to existing structures, ten days after casting the concrete cores (which simulate herein an existing column), the UHP-FRCC jackets are cast around using paper cylinders with a height of 178 mm as a disposable formwork.

5

Reviewer’s comments

6 In Table 4, the four types of thickness of the UHP-FRCC jacket should be tested for all series, not just in FA20. Because, the equations (1) and (2) are obtained from these test data, the precision of the relationship for equations (1) and (2) can be ensured in the presence of enough data.

Authors’ answer

We agree with the point of view of the reviewer: the larger the number of test, the better the precision of the relationships proposed herein. However, the results of the 33 tests reported in this paper seems to be sufficient to validate both Eq.(1) and Eq.(2).

6

Reviewer’s comments

7 The flowability of all series should be provided. In general, the particle size of fly ash is not same with that of the cement, which can be influenced the flowability of UHP-FRCC.

Authors’ answer

All results of the flow tests are now reported in the manuscript at:

Page 3, lines 29-33

Flow table tests were performed   in accordance with the Japanese Standard JIS R 5201 [20], which complies with ASTM C 1437 [21]. In particular, for FA0, FA20, FA50 and FA70 a diameter of 190 mm, 260 mm, 290 mm, and 250 mm, was respectively measured in the tests. Having these consistencies, all the UHP-FRCs were cast and compacted in the jacket mould without any segregation.

7

Reviewer’s comments

8 In section 3.1, the brief analysis about the Young modulus and the Poisson’s ratio should be added.

Authors’ answer

As suggested by the reviewer, the following lines report a brief comment on the Young modulus and the Poisson’s ratio

Page 5, lines 15-16

Ecm increased with the thickness of the jacket due to the confined effect. In the same manner, n decreased as the jacket thickness increased.

9

Reviewer’s comments

9 In Figure 5, the goodness of fit between test results and trend lines should be provided.

Authors’ answer

According to the reviewer’s requirement, Fig.5b now reports the values of s and R2 for each trend line.

10

Reviewer’s comments

10 In Figure 8, please provide more evidences and discussions about the defined possible best solutions and the dashed lines. The dashed line between 37.5 mm-75 mm in FA20 seemed also falls within Zone 3.

Authors’ answer

We completely agree with the point of view of the reviewer. Accordingly, Fig. 8 has been modified and the following sentence has been added at

Page 9, lines 12-14

The same is also valid for some thickness (within the range 25~37.5 mm) of the series FA20, as evidenced by the dashed line reported in Figure 8.

Reviewer 2 Report

The article deals with an interesting issue and the research results are worth publishing. However, important additions and clarifications are needed.

No data on the properties of the ingredients, this should be completed. No data on the properties of UHPC, this should be completed. Explain the adopted sample preparation procedure, especially in the aspect of heat treatment. What is its purpose? What is its impact on UHPC properties itself and jacket samples? Do the Authors include thermal treatment in the analysis of CO2 emissions? Did the authors investigate the contraction and crack resistance of UHPC? Did the cracks occur? It is important from durability point of view. How did the UHPC temperature change and what was its gradient? Has this aspect been the subject of research? This issue is important with the jacket structure and its durability. Authors should remember that the results obtained refer to specific components, especially when it comes to FA.

Author Response

1

Reviewer’s comments

No data on the properties of the ingredients, this should be completed. No data on the properties of UHPC, this should be completed.

Authors’ answer

The data regarding the components of concrete core are now reported at

Page 2, lines 29-33

According to the mix proportion shown in Table 1, the concrete cores have been made with High Early Strength Portland Cement (HESP; Density: 3.14 g/cm3, Specific surface area: 4490 cm2/g Ignition loss: 1.08%), the combination of land sand and crushed sand as fine aggregates (S), crushed stone as coarse aggregates (G), tap water (W), and superplasticizer (SP1; Polycarboxylate-based, Density: 1.03 g/cm3).

The data regarding the components of UHPC jackets are now reported at:

Page 3, lines 4-17

The UHP-FRCCs used herein are made by the following materials [3, 13-14]:

·         Low Heat Portland Cement (LHC; Density: 3.24 g/cm3, Specific surface area: 3640 cm2/g, C2S: 57%, C3A: 3%, MgO: 0.6%, SO3: 2.78%, Ignition loss: 0.72%)

·         Undensified Silica Fume (SF; Density: 2.20 g/cm3, Bulk density: 0.20 – 0.35 g/cm3, Coarse particles >45 µm: less than 1.5, SiO2: more than 90%, Ignition loss: less than 3.0%)

·         Fly Ash (FA; Density: 2.31 g/cm3, Specific surface area: 4050 cm2/g, Coarse particles >45 µm: 5%, SiO2: 54.8%, Ignition loss: 1.2%)

·         Silica sand (Ss; Density: 2.60 g/cm3, Average particle size: 0.212 mm, SiO2: 98.49%, Al2O3: 0.49%)

·         Wollastonite mineral fibers (Wo; CaSiO3, Density: 2.60 g/cm3, Length: 50 – 2000 μm, Ascpect ratio: 3 – 20, SiO2: 49.71%, CaO: 45.87%, Ignition loss: 1.94%)

·         Specific superplasticizer (SP2; Polycarboxylate-based, Density: 1.05 g/cm3)

·         De-foaming Agent (DA; Density: 1.05 g/cm3)

·         Steel micro-fibers (OL - 1% in volume), with a length of 6 mm (see Figure 2a)

·         Steel macro-fibers (HDR - 1.5% in volume), with a length of 30 mm (see Figure 2b).

2

Reviewer’s comments

Explain the adopted sample preparation procedure, especially in the aspect of heat treatment. What is its purpose? What is its impact on UHPC properties itself and jacket samples? Do the Authors include thermal treatment in the analysis of CO2 emissions?

How did the UHPC temperature change and what was its gradient? Has this aspect been the subject of research? This issue is important with the jacket structure and its durability. Authors should remember that the results obtained refer to specific components, especially when it comes to FA. 

Authors’ answer

The heat treatment is not the subject of this paper. However, according to the reviewer’s suggestions, the following lines have been introduced to justify the use of steam curing and avoiding the presence of a thermal gradient:

Page 4, lines 11-15

Then, the paper formworks were removed 2 days later and the specimens were subjected to steam curing for the following 48 hours. In this way, the hydration process of the binder was accelerated, especially in the UHP-FRCC jackets containing large amounts of fly ash. As depicted in Figure 3, the temperature was slowly increased at a rate of 15ºC per hour up to 90ºC to avoid crack generation produced by the thermal gradient.

Moreover, in the preparation of all the samples, the same procedure (illustrated in Fig.3) was adopted. As a comparative analysis among all the specimens is performed, the carbon footprint of the preparation process (including steam curing) is not taken into account. This aspect is reported at:

Page 8, lines 1-3

Here, the CO2 emission due to the steam curing is not taken into account, because all the series were subjected to the same procedures and, subsequently, a comparative analysis among all the specimens is performed.

3

Reviewer’s comments

Did the authors investigate the contraction and crack resistance of UHPC? Did the cracks occur? It is important from durability point of view.

Authors’ answer

To answer to the reviewer’s questions, the resistance of the UHPC has been added (see Table 6) and the growth of the tensile cracks in the UHPC jacket is also described at:

Page 5, lines 7-24

Table 5 shows the mechanical properties of the UHP-FRCC used in the jackets. Both of the compressive strength and the elastic modulus do not change as the age because of the acceleration of hydration due to the steam curing. The results of the uniaxial compression tests on the jacketed specimens are summarised in Table 5. In this table, the average values of the maximum load (Pmax) - and of the compressive strength (σmax) as well, the Young Modulus (Ecm) and the Poisson’s Ratio (n) of the composite specimens are reported. Both Ecm and n were calculated at one-third of the maximum load, according to the Japanese Standard JIS A 1149 [22], which complies with ISO 6784 [23]. In Table 6, the strength is also normalized with respect to the strength of the unconfined concrete cylinders (i.e., fc_CORE). Ecm increased with the thickness of the jacket due to the confined effect. In the same manner, n decreased as the jacket thickness increased. The tests on the specimen shown in Figure 1 were performed until the failure, which is generally produced by the formation of a large tensile crack in the UHP-FRCC jacket (see the right edge photo of Figure 3). During the first stage of loading, the jacked was uncracked and a linear relationship between the stress (calculated by dividing the load by the cross-sectional area of the core concrete) and the strain can be observed (see Figure 4). When the first crack appeared in the jacket, the slope of this relationship drastically reduces in all the specimens. This is due to the fact that multiple fine cracks, having a width lower than 0.1 mm, formed in the jacket. Despite the growing number of cracks, the stress continuously increased up to peak, where the tensile strains localised in a single crack of the jacket and the failure occurred.

Reviewer 3 Report

The paper entitled “Tailoring confining jacket for concrete column using ultra high performance-fiber reinforced cementitious composites (UHP-FRCC) with high volume fly ash (HVFA)” investigates the effect of cement replacement in ultra-high performance fibre reinforced composite paste. The research is interesting and the results are convincing but to further improve the manuscript, the following comments need to be addressed.

Abstract is well written but I suggest to include quantitative results too.

Introduction needs an improvement, particularly in ecological aspects (CO2 emissions)

Apparently the water to binder ratio of 0.16 was used in this study which is not enough for flowability. What was the ground for choosing such a low water content?

Is there any information on the compressive performance of samples at lower ages (7 and/or 14 days)? This is important as it gives an insight on how the jacket contributing to the performance of final product.

Why steam curing was performed starting at 3 days and why for 48 hours?

It would be preferable to present the compressive results in a graph rather than table so it is easier to compare the results. Also STD or SE must be added.

Please provide citation for Table 6’s caption

Author contribution and Funding need to be completed.

The paper is well organised but it lacks of a good discussion. The authors only presented the data without critical explanation on why such a results have been obtained or even comparing the results with the similar research studies. I suggest the authors carefully address this comment.

Author Response

1

Reviewer’s comments

Abstract is well written but I suggest to include quantitative results too.

Authors’ answer

According to the editors’ suggestion, the following sentences are included in the abstract

Page 1, lines 22-24

As a result, a replacement of about 50 % of cement with fly ash, and a suitable thickness of the UHP-FRCC jacket, may ensure the lowest environmental impact without compromising the mechanical performances.

2

Reviewer’s comments

Introduction needs an improvement, particularly in ecological aspects (COemissions)

Authors’ answer

According to the reviewer’s requirement, the following lines have been added to the introduction:

Page 1, line 43 and pag.2, lines 1-4

Indeed, it is unanimously accepted that the compressive strength of concrete is in direct proportion with the power 2 of the cement content and, consequently, this strength is proportional to the power 2 of the CO2 emission per cubic meter of concrete. Therefore, the “material substitution strategy” introduced by Habert and Roussel [11] is generally adopted.

3

Reviewer’s comments

Apparently the water to binder ratio of 0.16 was used in this study which is not enough for flowability. What was the ground for choosing such a low water content?

Why steam curing was performed starting at 3 days and why for 48 hours?

Authors’ answer

The definition of the optimal water/binder ratio of UHPFRCC, and of the steam curing as well, are out of the scope of this paper. They are obtained in previous analyses, as specified at:

Page 3, lines 26-28

In each series, the water-binder ratio is constant and equal to 0.16, because it guarantees a good balance between the flowability properties and the strength of the hardened concrete, according to the previous study of the authors [3].

4

Reviewer’s comments

Is there any information on the compressive performance of samples at lower ages (7 and/or 14 days)? This is important as it gives an insight on how the jacket contributing to the performance of final product.

Authors’ answer

The compressive strengths of UHP-FRCCs at 7 and 28 days are now reported in Table 5

5

Reviewer’s comments

It would be preferable to present the compressive results in a graph rather than table so it is easier to compare the results. Also STD or SE must be added.

Authors’ answer

Some of the stress-diagrams of the compression tests are reported in Fig.4. In this figure, it is possible to observe both the effect of cement substitution with the same thickness of the layer (Fig.4a), and the effect produced by different layers with the same content of cement (Fig.4b). In the opinion of the authors, other diagrams are not necessary, because, from a design point of view, it is sufficient the maximum stress reported in Table 6. Also, the values of STD seem unnecessary, as they have to be computed on three values.

6

Reviewer’s comments

Please provide citation for Table 6’s caption.

Authors’ answer

According to the reviewer’s suggestion, paper [20] is now cited in the caption of Fig.7 (the former Figure 6)

7

Reviewer’s comments

Author contribution and Funding need to be completed.

Authors’ answer

Both the Author contribution and Funding are now complete, as suggested by the reviewer.

8

Reviewer’s comments

The paper is well organized but it lacks of a good discussion. The authors only presented the data without critical explanation on why such results have been obtained or even comparing the results with the similar research studies. I suggest the authors carefully address this comment.

Authors’ answer

Some comments on the results have been introduced at:

Page 5, lines 16-24

The tests on the specimen shown in Fig.1 were performed until the failure, which is generally produced by the formation of a large tensile crack in the UHP-FRCC jacket (see the right edge photo of Figure 3). During the first stage of loading, the jacked was uncracked and a linear relationship between the stress (calculated by dividing the load by the cross-sectional area of the core concrete) and the strain can be observed (see Figure 4). When the first crack appeared in the jacket, the slope of this relationship drastically reduces in all the specimens. This is due to the fact that multiple fine cracks, having a width lower than 0.1 mm, formed in the jacket. Despite the growing number of cracks, the stress continuously increased up to peak, where the tensile strains localised in a single crack of the jacket and the failure occurred.

Round 2

Reviewer 1 Report

The content is well revised.

Reviewer 2 Report

Accept in present form